# Zipime-Weka-Schista study protocol: a longitudinal cohort study and economic evaluation of an integrated home-based approach for genital multipathogen screening in women, including female genital schistosomiasis, human papillomavirus, Trichomonas and HIV in Zambia

Kwame Shanaube,[1] Rhoda Ndubani,[1] Helen Kelly [iD],[2] Emily Webb,[3] Philippe Mayaud,[2] Olimpia Lamberti,[2] Jennifer Fitzpatrick,[1] Nkatya Kasese,[1] Amy Sturt,[4] Lisette Van Lieshout,[5] Govert Van Dam,[5] Paul L A M Corstjens,[6] Barry Kosloff,[1,7] Virginia Bond,[1,8] Richard Hayes,[3] Fern Terris-Prestholt,[9] Bonnie Webster,[10] Bellington Vwalika,[11] Isaiah Hansingo,[12] Helen Ayles,[1,2] Amaya L Bustinduy [iD] [2]

For numbered affiliations see end of article.

**Correspondence to**
Dr Amaya L Bustinduy;
Amaya.Bustinduy@lshtm.ac.uk

## ABSTRACT

**Introduction** Multiplathogen home-based self-sampling offers an opportunity to increase access to screening and treatment in endemic settings with high coinfection prevalence of sexually transmitted (HIV, *Trichomonas vaginalis (Tv),* human papillomavirus (HPV)) and non-sexually transmitted pathogens (*Schistosoma haematobium (Sh)*). Chronic coinfections may lead to disability (female genital schistosomiasis) and death (cervical cancer). The Zipime-Weka-Schista (Do self-testing sister!) study aims to evaluate the validity, acceptability, uptake, impact and cost-effectiveness of multipathogen self-sampling for genital infections among women in Zambia.

**Methods and analysis** This is a longitudinal cohort study aiming to enrol 2500 non-pregnant, sexually active and non-menstruating women aged 15–50 years from two districts in Zambia with 2-year follow-up. During home visits, community health workers offer HIV and *Tv* self-testing and cervicovaginal self-swabs for (1) HPV by GeneXpert and, (2) *Sh* DNA detection by conventional (PCR)and isothermal (recombinase polymerase assay) molecular methods. *Schistosoma* ova and circulating anodic antigen are detected in urine. At a clinic follow-up, midwives perform the same procedures and obtain hand-held colposcopic images. High-risk HPV positive women are referred for a two-quadrant cervical biopsy according to age and HIV status. A cost-effectiveness analysis is conducted in parallel.

**Ethics and dissemination** The University of Zambia Biomedical Research Ethics Committee (UNZABREC) (reference: 1858-2021), the London School of Hygiene and Tropical Medicine (reference: 25258), Ministry of Health and local superintendents approved the study in September 2021.Written informed consent was obtained from all participants prior to enrolment. Identifiable data collected are stored securely and their confidentiality is protected in accordance with the Data Protection Act 1998.

## STRENGTHS AND LIMITATIONS OF THIS STUDY

⇒ The longitudinal design of the study will allow measurements of incidence and persistence of human papillomavirus (HPV) in association with female genital schistosomiasis (FGS) and HIV and its joint contribution to cervical precancer and cancer.

⇒ The one-stop home self-sampling (FGS, high-risk HPV) and self-testing (HIV and Trichomonas) approach, if acceptable by participants, can become a scalable integrated screening strategy.

⇒ Cost-effectiveness data from the study will inform of community-based integration of FGS within the wider sexual and reproductive health screening in Zambia.

⇒ Recruitment may be limited by stigma pertaining to coexisting sexually transmitted infections.

⇒ Results may be delayed due to the limited availability of histopathology laboratories.

## INTRODUCTION

It is estimated that around 45 million women living in sub-Sahara Africa (SSA)

are affected by female genital schistosomiasis (FGS), a chronic, neglected and disabling gynaecological disorder associated with infertility, dyspareunia and symptoms mimicking sexually transmitted infections (STIs).[1 2] FGS is difficult to diagnose and is part of the wider spectrum of urogenital disease caused by the waterborne parasite *Schistosoma haematobium* (*Sh*). In Zambia, reported FGS prevalence ranges from 33% to 75% of those with urinary schistosomiasis in endemic areas, suggesting a potentially high FGS burden. Data are not yet available on the FGS burden in the general population of Zambia. Furthermore, awareness of the disease and non-sexual mode of transmission is largely absent in endemic communities where information and education are urgently needed.[3]

Conventional FGS diagnosis is challenging as it relies on costly equipment and high-level specialised training which is seldom available in resource-limited countries. Therefore, accurate estimation of disease burden is hindered. However, the past decade has seen a dramatic increase in knowledge generation on many facets of FGS, and particularly on diagnostic strategies closer to the point of care.[4] Studies have shown acceptable sensitivity of genital swabs for the DNA detection of *Sh* from women's genital tracts, potentially offering, for the first time, a scalable diagnostic approach for a parasitic disease that has been unacceptably neglected.[3 4] Genital self-swabs are well accepted by participants and are known to increase uptake across settings.[5 6]

There is a growing evidence of an association of FGS with prevalent HIV, and suggestion of increased incidence and transmission of HIV in the presence of schistosome infections in women.[7] The lesions and tissue damage caused by FGS can provide an easy route for STI infections.[8] Further, women with schistosome worm infections and with FGS are potentially far more susceptible and at an increased risk of becoming HIV positive.[9] However, there is still a knowledge vacuum on the role of STIs in HIV acquisition in the presence of FGS, despite both STIs and *S. haematobium* infection being shown to be independently associated with HIV acquisition.[8 10 11]

There have been small studies suggesting an increased prevalence of high-risk (HR-) human papillomavirus (HPV), the aetiological agent of cervical cancer, in women with FGS compared with those without[12], higher odds of having an abnormal visual examination with acetic acid (VIA)[13] and some suggestion of potential links between FGS and cervical cancer.[14 15] In Zambia, cervical cancer is the most common female cancer, particularly in women aged 15–44 years[16 17] and with high associated mortality (43.4 per 100 000 women). The lack of accessible screening and treatment programmes[18] and HPV vaccination contribute to the high cervical cancer incidence in these settings.[19] HPV-DNA-based tests are recommended for primary screening of cervical cancer as part of the WHO 2021 cervical cancer screening and treatment guidelines,[20] as better strategies than visual diagnosis.[21 22] Despite validated self-sampling strategies for the detection of oncogenic or HR-HPV genotypes,

community-based screening is not yet widely adopted alone or integrated with screening for other pathogens due to the high cost and infrastructure requirements of currently available tests.[23]

This study protocol describes the development and costing of a community-based comprehensive package for SRH screening in Zambia, including STIs (HPV, HIV, *Trichomonas vaginalis* (*Tv*)) and non-STIs (*Sh*) in different endemicity settings with the long-term aim of preventing life-threatening and disabling conditions such as cervical cancer, HIV and FGS. The overall justification of this study lies in the dearth of research exploring the feasibility of integrated screening and testing strategies for genital coinfections, and the epidemiological associations between them.[24–26]

## METHODS AND DESIGN
### Study design
The Zipime Weka Schista (Do self-testing sister!) study is a longitudinal cohort designed as a holistic approach to FGS detection at scale in the community. The study design will test the hypothesis that integration of home self-sampling for the screening of FGS in conjunction with HR-HPV and self-testing for HIV and STIs is a diagnostically accurate, cost-effective and self-empowering strategy that will increase the detection of cases and improve access to care for women of reproductive age in SSA.

The two main aims of the Zipime Weka Schista study are:

1. To assess the performance of genital self-sampling, compared with healthcare provider sampling obtained for the detection of *Sh* by molecular testing (real-time PCR and recombinase polymerase assay (RPA)) across two different *Sh* endemicity settings in Zambia.
2. To assess the observed costs of integrating community-based multiple pathogen genital self-sampling (*Sh*, HPV, STIs) and oral self-test (HIV) into a one-stop home intervention and project costs at scale.

Our secondary aims are to explore the epidemiological and immunological interrelationships between FGS and HPV as a potential risk factor for cervical cancer and expand knowledge on HIV acquisition in the presence of coinfections. We will also measure acceptability and feasibility of home-based multiple genital self-sampling and to measure HIV linkage to care and cervical cancer management after home self-sampling and testing. We will further aim to evaluate the effect of FGS and other STI on the diagnostic accuracy of visual inspection for the detection of cervical precancer (CIN2+, CIN3+) and to describe the association of FGS, HPV, HIV and STIs with the presence concentration, diversity and relative abundance of other key cervicovaginal microbiota and infections. We will also aim to consolidate testing for FGS and HPV into one field-applicable multiplex molecular assay and to recruit a subset of male household members of recruited participants screening male genital schistosomiasis.

## Study setting and population

The study is being conducted in Lusaka and Southern provinces of Zambia. The specific study study areas are two sites in Kafue district with high (estimated *Sh* prevalence >70%) and moderate (estimated *Sh* prevalence >50%) prevalence and another site in Livingstone district with low prevalence (estimated *Sh* prevalence <10%). Each woman will have one home visit and one clinic visit at baseline with annual follow-up for 2 years. Enrolment of eligible women began in January 2022 and is expected to be completed in May 2024.

## Inclusion/exclusion criteria

All non-pregnant, sexually active women aged 15–50 years who are currently not menstruating at the time of recruitment and sampling and are able to provide consent are invited to take part in the study. In addition, women should be Zambian residents of the endemic study area for at least 1 year. Women are excluded prior to their sexual debut, pregnant or currently menstruating at the time of recruitment and sampling.

## Sampling strategy and data collection procedures

Community-based cluster sampling was used to obtain a random sample of participants from the study communities. Communities and their households were mapped and each community was subdivided into blocks of approximately 50 households; each one would yield approximately 30 enrolled women. These community blocks were the primary sampling units, with all households in a selected block visited, and all eligible women aged 15–50 years invited to take part. Blocks were randomly selected using random number generation in Stata 18 and are being visited sequentially in the order that they were randomly selected. Participating women are assigned a unique identifier. Field data are entered using the London School of Hygiene and Tropical Medicine (LSHTM) ODK system and uploaded to a secure server in Zambia. The data are then transferred to an LSHTM secure server. The chief investigator is responsible for the data analysis with the support of the Department of Infectious Diseases Epidemiology at LSHTM.

## Sample size and study power

A total of 2500 women will be randomly selected from 2 districts (1250 from each district). Based on previous assessments of FGS using PCR testing of genital samples from women in the study setting, we assume that 5%–10% of genital samples from healthcare provider sampling will have *Sh* detected by molecular testing (5% in the lower endemicity area, 10% in the higher endemicity area). Taking clinic-based sampling as the gold standard, allowing for 10% loss to follow-up between the home and clinic sampling, and assuming that the diagnostic sensitivity and specificity of self-sampling compared with clinic-based sampling for the detection of *Sh* are 70% and 85%, respectively, we will be able to estimate the assumed target diagnostic sensitivity of 70% with a 95% CI of 62.3% to

76.6% overall, and the assumed target diagnostic specificity of 85% with a 95% CI of 83.4% to 86.5% (using the Clopper-Pearson formula to calculate an exact 95% CI for this proportion). Considering individual districts, we will be able to estimate a sensitivity of 70% with a 95% CI of 60.6% to 78.2% in the higher endemicity area and a 95% CI of 55.9% to 81.2% in the lower endemicity area. Corresponding CIs for specificity would be 83%–87% in each area.

## Field procedures

Community sensitisation activities take place to inform target communities of the study's aims and promote voluntary participation. Different support materials are used. Some of these include local drama groups, information brochures and an FGS manual edited by Zambart. Study community workers (SCWs) register expressions of interest and conduct home visits within the randomised zones to women that have expressed interest in participating in the study. All SCWs are females. Every participant has a total of three visits during the study: baseline, 12 and 24 months from recruitment. A schematic study flow is shown in figure 1.

### Home visits

Home visits are conducted by SCWs who perform a demonstration of self-sampling, using a 3D model for illustration (online supplemental figure 1). SCWs also obtain consent from participants and administer demographic and exposure questionnaires. After obtaining informed consent and completing questionnaires on demography, clinical symptoms and other relevant health and water contact information, the SCWs offer participants two genital self-swabs. The first swab is stored in PrimeStore MTM molecular transport media (donated by Longhorn Vaccines and Diagnostics, Bethesda, Maryland, USA) for the detection of *Sh* by molecular methods (PCR and RPA). The second swab is stored in Cytopreserve for HR-HPV detection by Xpert.[27] Participants are also offered two commercial self-tests: one oral (saliva) for HIV[28] and one vaginal self-swab for *Tv*.[29] For both self-tests, results are read by the SCW and given immediately to the participant. After all the self-sampling swabs are delivered, the SCW record acceptability results based on a Likert Scale. A confirmatory test is performed on site if there is a positive self-test for HIV. All HIV-positive women are referred to the local clinic for further management and linkage to care. For *Tv*, women are referred for treatment at the clinic visit. A urine sample is analysed via microscopy for *Sh* ova detection and circulating anodic antigen (CAA).[30]

### Clinic visit

A follow-up visit takes place at a local clinic. A trained healthcare provider (midwife) collects two cervicovaginal swabs and 5 mL of cervicovaginal lavage (CVL). Swabs are placed in PrimeStore MTM molecular transport media (donated by Longhorn Vaccines and Diagnostics) and

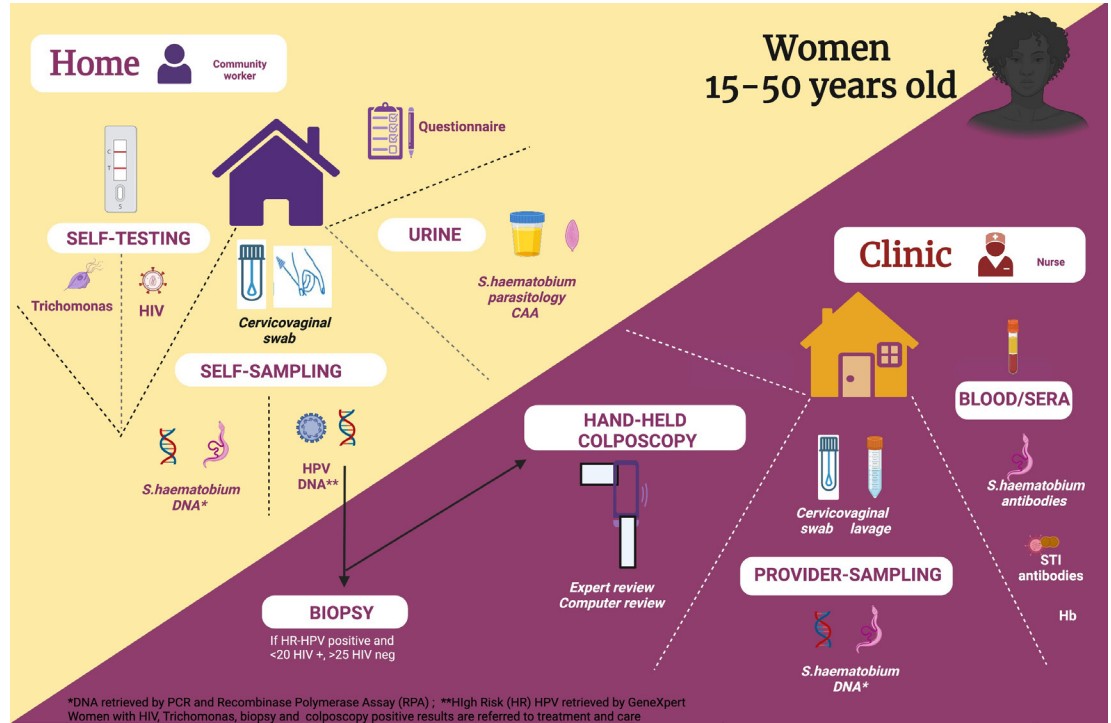

**Figure 1** Schista study flow. CAA, circulating anodic antigen; HPV, human papillomavirus; STIs, sexually transmitted infections.

CVL is stored at −20°C on site for future detection of: (1) *Sh* DNA by real-time PCR as the reference assay for FGS[31]; (2) *Sh* DNA by Xpert CT/NG (Chlamydia Trachomatis/Neisseria gonorrhoeae)[32] and studies of biomarkers and cervivovaginal microbiome. Approximately 5 mL of blood is collected by venipuncture in an EDTA tube for the detection of *Schistosoma* and STI serology. A hand-held colposcope, with an enhanced visual assessment system, is used to image the participant's cervix and vagina (Mobil-eODT, Tel Aviv, Israel). Praziquantel and metronidazole are available free of charge for study partcipants with any positive result.[33]

### Biopsy procedure

Women testing positive for any HR HPV by PCR (Xpert HPV) during the home visit and depending on their age and HIV status (see below) are referred for colposcopy. Visual inspection using acetic acid and lugol's iodine (VIA/VILI) is performed by a trained gynaecologist before biopsy of visible lesions. If no lesions are visible, two random biopsies are taken from normal areas including the transformation zone (at 12 and 6 o'clock) (figure 2). Cervical biopsies will be processed at the pathology laboratory at National Health laboratory Service, Johannesburg, South Africa and read using the two-tier CIN classification system.

HIV-negative women under the age of 25 and women living with HIV under the age of 20 years have all procedures performed, including HPV testing in the community, but they do not undergo biopsy in clinic despite positive HR-HPV results.[34] This strategy follows ethical concerns of unnecessary biopsy and management of

cervical lesions based on known HPV clearance and low risk of cervical precancer in young women.[35] For these women, there is annual longitudinal follow-up for 2 years with repeated infection and clinical reassessments.

### Women diagnosed with cervical precancer and invasive cervical cancer

If found to have histopathology verified CIN2+, participants are recalled for CIN2+management. This visit is scheduled at the earliest convenient date once the result

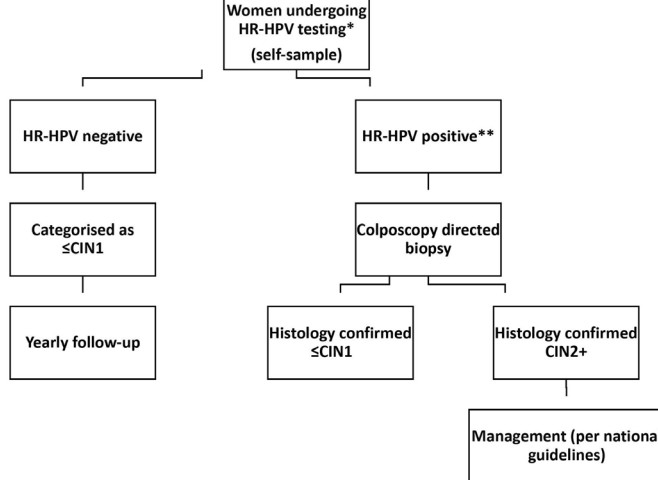

**Figure 2** Flow chart for cervical precancer determination in HIV negative an women living with HIV (WLHIV). *HR-HPV detected using Xpert; **HR-HPV positive WLHIV aged 20 years and older and HIV-negative women aged 25 years and older only. HR-HPV, high-risk human papillomavirus.

**Table 1** Sample storage and processing for the Schista study

| Sample | Location | Storage | Sample processing | Location of processing |
|---|---|---|---|---|
| Self-collected swab Provider-collected swab | Home | Room temperature (Stored in Primestore media) | Sh DNA (PCR, RPA) GeneXpert- High Risk HPV DNA* | Zambart (Zambia) |
| Urine for microscopy | Home | Room temperature | Sh eggs Urine haemastix | District Laboratory (Kafue, Livingstone, Zambia) |
| Urine stored for CAA | Clinic | 4°C | Circulating Anodic Antigen | Leiden University Medical Centre (The Netherlands) |
| Cervicovaginal lavage | Clinic | −20°C | Sh DNA (PCR, RPA) GeneXpert- CT/NG Microbiota | Zambart (Zambia) |
| Blood for serology | Clinic | −20°C | Sh, STI antibody | Zambart (Zambia) |
| Colposcopy images | Clinic | Encrypted server | Expert review computer analysis | Zambia London School of Hygien and Tropical Medicine (UK) |

*High-risk HPV by self-collected swab only.
CAA, circulating anodic antigen; HPV, human papillomavirus; RPA, recombinase polymerase assay; *Sh*, *Schistosoma haematobium*; STIs, sexually transmitted infections.

is known. In Zambia, all patients with cancer are referred to the Cancer Diseases Hospital (CDH) based in Lusaka. Patients are filtered through the district and provincial hospitals serving as primary centres for cervical cancer screening countrywide before being referred to CDH for definitive treatment. Treatment at CDH is done according to the nationally and globally developed guidelines.[36]

### Laboratory procedures

The collection, storage and processing of all study samples are summarised in table 1. Urine samples are processed for (1) urine parasitology: approximately 10 mL (minimum) of urine is processed by centrifugation to detect Sh eggs by microscopy performed at the local government clinics in Kafue and Livingstone in Zambia by a trained technician and (2) CAA antigen assay: a 2 mL of urine is stored at 4°C and shipped to Lusaka and further to Leiden University Medical Center, The Netherlands.[37]

All genital samples (home/clinic) are processed for (1) Detection of schistosoma DNA using ITS-2 real-time PCR processed at Zambart laboratories in Lusaka, Zambia.[31 38] From the DNA already extracted for PCR additional well-validated qPCR assays will be used[39] to characterise the vaginal microbiota by quantifying key species and (2) Detection of Schistosoma DNA using RPA. Up to 2 mL of fluid from vaginal lavage is stored at −20°C and shipped to Lusaka. This assay is processed at Zambart laboratory in Zambia[40 41] and (3) Inflammatory Markers. Up to 3 mL of fluid from vaginal lavage is stored at −80°C and shipped to Lusaka.[25 31 42]

### Development of a multiplex assay for *Sh* and HR-HPV testing

In a exploratory study, we will aim to integrate into one molecular assay the RPA for *Sh* and HPV to increase accessibility and feasibility of molecular dual pathogen screenings after encouraging results on the validaty of the CVL RPA assay against standard real-time PCR,[43] which could be coupled with oncogenic HPV genotypes.[44]

### Qualitative assessment

#### In-depth feasibility and acceptability of integrated sampling, testing and treatment

Previous work has documented the acceptability of performing multiple genital swabs at a single encounter.[27] For the current proposal, in-depth cognitive interviews will be delivered to a subset of participants to evaluate the acceptability of the multipathogen self-swabbing strategy (three genital swabs and one oral swab).[45]

### Statistical analysis

To assess the performance of genital self-sampling compared with healthcare provider sampling for the detection of *Sh*, sensitivity, specificity, positive predictive value (PPV) and negative predictive value (NPV) will be calculated together with 95% CIs. Results will be presented overall and stratified by geographical setting. To explore associations of baseline FGS with the prevalence of cervical precancer at baseline and cervical precancer incidence at 12 and 24 months, logistic regression will be used to estimate ORs. Associations between baseline FGS and HR-HPV persistence at 12 months will be estimated with generalised estimating equation to account for multiple HR-HPV infection and multiple infection states (persistence and clearance).[46] The association of FGS with HIV acquisition will be assessed using Poisson regression, controlling for key confounders.

Other outcome measures obtained in the study will include (1) HIV incidence rate, HR-HPV clearance and persistence, prevalence and incidence of cervical precancer (histology confirmed CIN2 or CIN3+), measures of association between different exposures

(reproductive tract infections); (2) acceptability of multiple genital self-sampling interventions; (3) rates of antiretroviral initiation, continuation of treatment and access to care; (4) cervical precancer diagnosis and treatment, testing and management of STIs; (5) the presence or absence of cervicovaginal microbiota in women with FGS, HPV and STIs. Expressed as relative abundance (percentage abundance of each species), genomic concentration per micro-organism (genome equivalents per mL) and (6) sensitivity, specificity, PPV, NPV of visual inspection of the cervix according to FGS status and integrated molecular field-applicable testing platform for *Sh* and HR-HPV.

### Quality control

Several procedures have been employed to ensure the quality of the study data, maximising the validity and reliability of the data obtained and of the evaluation of the results: These include the following (a) 10% of all samples (positive and negative results from diagnostic assays) will be reviewed blindly by experts at different institutions; parasitology (Liverpool School of Tropical Medicine, Liverpool, UK), PCR (Natural History Museum, London, UK), colposcopic images (University of Zambia, Lusaka, Zambia).

### Economic analysis

A full economic costing from a provider's and patient's perspective is ongoing alongside data collection. Cost data are collected by combining a bottom-up and top-down approach following established costing guidelines,[47] and STI self-testing economic evaluations in the region. This allows us to estimate the total intervention cost, and unit costs per woman screened, FGS infection detected, per case of CIN2+ detected, per person linked to care, per person successfully treated overall and per pathogen identified. We will model the costs across a set of scale-up scenarios to understand the budget implications of introducing widescale self-screening. Populating a health economics decision model with the study outcomes, we will identify cost thresholds under which joint self-sampling for FGS, HPV, STIs and HIV is cost-effective. This will be relevant for informing the maximum threshold diagnostic kit price based on its potential cost-effectiveness.[48] MoH and policy-makers will be informed on the most cost-effective screening strategy to implement for FGS screening. This study constitutes the first economic evaluation of FGS.

### Patient and public involvement

Participants who are being recruited for this study did not take part in the design or implementation of the study. They are, however, deeply involved finding solutions for better follow-up through focus groups discussions and mobilisation activities in the community.

## DISCUSSION

Sexual and reproductive health is a critical area of development highlighted in the United Nations Sustainable Development Goal 3. There are an estimated 40 million women unknowingly living with FGS, with dire consequences for their sexual and reproductive lives which include infertility, pain with sexual relations, ectopic pregnancies and increased risk of HIV acquisition and cervical precancer development, mostly in countries with high prevalence of STIs. This study will provide high-quality evidence on the validity, acceptability, uptake and costs of integrating community-based multiple-pathogen genital self-sampling (Sh, HR-HPV, STIs) and oral self-test (HIV) into one-stop intervention. Findings will help inform policy and practice at local and regional levels. Importantly, this study brings together existing and new partnerships for a multidisciplinary investigative approach, to explore the value of integration of reproductive health screening practices through self-sampling and testing. The baseline results will provide further validation of self-swabs as an acceptable sampling strategy for FGS diagnosis in a larger sample population and with a wider scope than previously performed. Its longitudinal approach will also provide an allied impact assessment on the incidence of cervical precancer, HIV and STIs and early HIV detection and access to treatment for the wider improvement of women's SRH in Zambia. This is important within the growing evidence of a link between FGS and cervical cancer.[15] The clear interplay between genital infections highlights the urgent need to bolster programmes that jointly integrate strategies to screen and detect multiple genital infections with simple, affordable and as close to the point of care as possible.

Genital self-sampling and self-testing through community-based programmes are widely accepted strategies for the detection of devastating diseases like HIV in both adults and adolescents.[8 31 49] HIV self-testing programmes have been shown to be cost-effective. Genital self-sampling has enhanced access to health services among hard-to-reach populations such as young people,[31] and those who do not regularly access health screening services.[1 50] It also empowers women and protect their privacy since self-swabs can be performed at home or in other private places and increases compliance with testing. A high proportion of women, including those from resource-limited settings, have been found to prefer vaginal specimen self-collection[51] compared with clinic-based sampling. In addition to acceptability, two other factors make genital self-sampling advantageous: (1) the availability of vaginal self-sampling is effective for improving participation in specific SRH screening programmes and (2) the sensitivity of PCR-based assays on self-collected specimens compares favourably with physician-performed sampling.[1]

The synergistic approach proposed in this study is justified by the known and suspected interactions of the different genital coinfections. FGS is associated with genital inflammation, friable blood vessels and contact

bleeding.[52–54] The disruptions of the mucosal surface in FGS may place women at risk for HIV, as has been found for several STIs.[55–57] Women with urogenital schistosomiasis had a 3–4 fold higher odds of prevalent HIV in cross-sectional studies in Zimbabwe, Tanzania and Mozambique.[10 58–61] However, FGS as a risk factor for HIV is not yet integrated into ongoing screening programmes, including HIV self-testing strategies.[62–64]

Current FGS diagnostic strategies are limited because they require expertise and equipment that may not be readily available in low-income settings. Several diagnostics will be tested in this study to provide 'real-world' data on the scalability of the different approaches. Hand-held colposcopy as a low-cost device has been used in both cervical cancer screening and FGS studies.[4 65] However, expert review is still required for the identification of characteristic FGS lesions and inter-rater correlation is poor.[66] This study will enhance diagnostics by applying multipathogen DNA testing with conventional molecular assays that are field deployable, such as the RPA.[32 40] Following on the positive correlation between RPA assay from CVL against standard real-time PCR,[67] in the Schista study, we will aim to develop a multiplex RPA assay for Sh and HR-HPV to increase accessibility and feasibility of dual pathogen molecular screenings.

To make interventions ultimately sustainable, these need to be cost-effective. In this study, we aim to assess for the first time the observed costs of integrating community-based multiple-pathogen genital self-sampling (Sh, HPV, STIs) and oral self-test (HIV) in a one-stop intervention and project costs at scale for the justification of the diagnostic kit price based on its impact on disease prevalence.[26]

To conclude, the Zipime Weka Schista study has a number of potential benefits at both individual and community levels including diagnosis and treatment of previously undiagnosed Sh infection as well as diagnosis, treatment and linkage to care for cervical cancer, HIV and STIs. Multipathogen self-sampling and testing constitutes an unmissable opportunity to offer potentially life-changing health services closer to the users.[6]

**Author affiliations**
[1]Zambart, Lusaka, Zambia
[2]Department of Clinical Research, London School of Hygiene and Tropical Medicine, London, UK
[3]Epidemiology and Population Health, London School of Hygiene and Tropical Medicine, London, UK
[4]Department of Infectious Diseases, Veterans Affairs Health Care System, Palo Alto, UK
[5]Leiden University Medical Center, Leiden, The Netherlands
[6]Department of Cell and Chemical Biology, Leiden University Medical Center, Leiden, The Netherlands
[7]Longhorn Vaccines & Diagnostics, Bethesda, Maryland, USA
[8]London School of Hygiene & Tropical Medicine Centre of Global Change and Health, London, UK
[9]London School of Hygiene & Tropical Medicine, London, UK
[10]Natural History Museum, London, UK
[11]Department of Obstetrics & Gynaecology, University of Zambia School of Medicine, Lusaka, Zambia
[12]Gynecology, Livingstone Central Hospital, Livingstone, Zambia

**Acknowledgements** We would like to thank and acknowledge all the participants, Schista! study fieldworkers and midwives and the wider Schista! Study team who contributed towards the design of the study. Also, our partners in Zambia including the Ministry of Health and District Health Management Teams; the administrative and support teams at Zambart and LSHTM.

**Contributors** AB originally conceived and designed the study in consultation with PM, HA, BW, IH, FT-P and RH. KS wrote the first draft of this manuscript. BW, HK and AS advised on primary and some secondary objectives, FT-P and OL conceived the economic analysis, EW and NK helped design the statistical framework of the study and contributed to all drafts of the manuscript. BK, BW, LVL, JF, GVD and PLAMC contributed to the laboratory component of the study. RN, VB and BV contributed to the design of secondary objectives. All authors read and approved the final manuscript.

**Funding** This study is funded through a UKRI Future Leaders Fellowship (MR/T041900/1) awarded to Professor AL Bustinduy. Professor E Webb and Prof R Hayes received funding from MRC Grant Reference MR/K012126/1. This award is jointly funded by the UK Medical Research Council (MRC) and the UK Department for International Development (DFID) under the MRC/DFID Concordat agreement and is also part of the EDCTP2 programme supported by the European Union.

**Disclaimer** Funders had no role in study design, data collection and analysis, decision to publish, or preparation of the manuscript.

**Competing interests** None declared.

**Patient and public involvement** Patients and/or the public were not involved in the design, or conduct, or reporting, or dissemination plans of this research.

**Patient consent for publication** Not applicable.

**Provenance and peer review** Not commissioned; externally peer reviewed.

**ORCID iDs**
Helen Kelly http://orcid.org/0000-0001-5547-3807
Amaya L Bustinduy http://orcid.org/0000-0001-6131-4159

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
