## [Reviewer comments · BMJ Open]

ARTICLE DETAILS

TITLE (PROVISIONAL)	The Zipime-Weka-Schista study protocol: a longitudinal cohort study and economic evaluation of an integrated home-based approach for genital multi-pathogen screening in women, including female genital schistosomiasis, Human Papilloma Virus, Trichomonas and HIV in Zambia
AUTHORS	Shanaube, Kwame; Ndubani, Rhoda; Kelly, Helen; Webb, Emily; Mayaud, Philippe; Lamberti, Olimpia; Fitzpatrick, Jennifer; Kasese, Nkatya; Sturt, Amy; Van Lieshout, Lisette; Van Dam, Govert; Corstjens, Paul; Kosloff, Barry; Bond, Virginia; Hayes, Richard; Terris-Prestholt, Fern; Webster, Bonnie; Vwalika, Bellington; Hansingo, Isaiah; Ayles, Helen; Bustinduy, Amaya

VERSION 1 – REVIEW

REVIEWER	Kelly-Hanku, Angela Univ Sydney
REVIEW RETURNED	24-Nov-2023

GENERAL COMMENTS	Thank you for the opportunity to review this protocol. Overall I think the idea of the research is interesting, but the protocol needs significant work. I would indeed even ask does it need to be published. My major concern is that there is inconsistency, at times it reads that its a protocol of what will be done, at other times it reads like a paper describing what has been done. I think the protocol needs careful revision with this in mind. Some more general comments. The protocol refers to STIs when in fact other than HIV only 1 STI is being tested for. I am unclear why you would test for all these infections and not also take a vaginal swab for CT/NG given its associated with important reproductive outcomes for women and this is a study among women of reproductive age. I would have liked to see how women with HIV and hrHPV are being treated - no clinical algorithm included. This is an important oversight. Field procedures - when does community sensitisation take place? Positive test for HIV. With HIV self-testing it cannot be positive but can be reactive. Please correct. How is confirmatory testing for HIV done on site?
---

	Why are cognitive interviews being done. This is usually something done by police. I think there is overreach re STIs throughout the protocol.
--	--

REVIEWER	Burrowes, Sahai Touro University California
REVIEW RETURNED	20-Dec-2023

GENERAL COMMENTS	I was happy to have the opportunity to review a manuscript on community-based schistosomiasis testing, as schistosomiasis causes great suffering but is woefully under-researched. The protocol is ambitious, but I find the approach of simultaneously testing for multiple reproductive tract infections to be sound as their symptoms overlap, and I have found that misdiagnosis between them is common. It is also encouraging to see a study of the relationship between HIV, HPV, and schistosomiasis infection. I have minor suggestions listed below, but overall, the protocol is methodologically sound, reasonably well-written, and well-organized. Editorial The language could be cleaned up throughout the document. Minor typographical and grammatical errors are scattered throughout the document, and it will need thorough copy-editing before publication. In this editorial review, I suggest reducing the use of numbered lists as they are largely unnecessary in the text, and I don't believe they are commonly used in journal articles. Methods  • Please state the formula used to calculate the study sample size and whether the clustering of responses by community block was considered when determining the sample size. • One of the study aims was to empower women, but there is no information provided about the types of education that women receive before/during their participation in the study or the training that the community health workers will receive to educate women about these illnesses. A few more sentences about this would be nice to have. Educating women about the fact that their genital tract symptoms could mean several different diseases is important for future care-seeking and for reducing stigma. • The economic cost-effectiveness analysis seemed like an afterthought as it is discussed only cursorily. And I wonder if it shouldn't be removed from this protocol and written up separately in more detail. This protocol has many study outcomes and analyses regarding the shisto, HPV, and HPV infection. Perhaps the space in this manuscript would be better used to discuss those outcomes in more detail. • Please cross-check the statistical analysis sections with the study design sections to ensure that the objectives/outcomes discussed are the same throughout the manuscript. For example, ART initiation is mentioned in the statistical analysis section but not as an outcome in the study design. Tables and Figures
---

	 • For Figure 2, I would suggest adding the actions taken for women who are HIV-negative and HR-HPV-positive to the flow chart so it is clear what follow-up care they receive. Are women who are HR-HPV positive but who are not HIV infected referred to for VIA or other further care? VIA for HIV+ women is not listed in Figure 2, but it is mentioned in the text. This should be clarified. • It's a bit repetitive, but I would have liked to have the location of sample collection included in Table 1. The location for the various sample collections is a little muddy in the text and would be easier to read in the table than in Figure 1. • Having a table or figure listing the various study objectives/outcome measures with their corresponding data collection and analysis methods and timing would be nice. To me, it seemed that study objectives were scattered throughout the document making it difficult to keep track of what outcomes the researchers would be measuring.
--	---

VERSION 1 – AUTHOR RESPONSE

Reviewer: 1

Dr. Angela Kelly-Hanku, Univ Sydney

Comments to the Author:

Thank you for the opportunity to review this protocol.

Overall I think the idea of the research is interesting, but the protocol needs significant work. I would indeed even ask does it need to be published.

My major concern is that there is inconsistency, at times it reads that its a protocol of what will be done, at other times it reads like a paper describing what has been done. I think the protocol needs careful revision with this in mind.

We thank Dr Kelly-Hanku for taking the time to review our protocol. To give some further background on our work, we have already published this protocol as a pre-print in MedRxiv and have had 150 downloads since October of 2023. There is great interest on our work because it is the first time FGS/HPV/HIV/Tv are included in an integrated approach to self-sampling at home. We strongly believe our published protocol can help researchers and implementers inform similar strategies going forward.

Some more general comments.

The protocol refers to STIs when in fact other than HIV only 1 STI is being tested for. I am unclear why you would test for all these infections and not also take a vaginal swab for CT/NG given its associated with important reproductive outcomes for women and this is a study among women of reproductive age.

Thank you for your comment. We are in fact testing for three STIs: HIV, HPV and Trichomonas.

These

STIs were chosen due to the known association with FGS through inflammation pathways promoting severe morbidity. Further, the longitudinal association between FGS and HPV (in women living with HIV) is important to ascertain to measure their joint impact on increased incidence of pre- and cervical cancer. (lines 145-156)

As the protocol is funded by a UKRI fellowship with limited budget, we had to prioritise spending and restrict the testing to what was scientifically more relevant as well as doable and affordable within the existing budget. Testing for CT/NG is indeed an interesting test to add on and has already been mentioned in the protocol's methodology (line 265). Testing will be considered from stored samples, if additional funding is obtained.

I would have liked to see how women with HIV and hrHPV are being treated - no clinical algorithm included. This is an important oversight.

We have devoted a large portion of our protocol to develop a clinical pathway of our participants who

are living with and without HIV. Depending on HIV status, women testing positive for HR-HPV follow an algorithm for further biopsy eligibility that is indeed included in the protocol- Figure 2. (Lines 273-280) After histopathology results are obtained, participants are referred to the national health service of Zambia for further pre-cancer and cancer care. (lines 288-294)

Field procedures - when does community sensitisation take place?

Lines 236-237. These are ongoing.

Positive test for HIV. With HIV self-testing it cannot be positive but can be reactive. Please correct.

How is confirmatory testing for HIV done on site?

Confirmatory HIV testing is done on site before referral to HIV clinic (lines 254-255)

Why are cognitive interviews being done. This is usually something done by police.

The cognitive interviews referred here are how in-depth interviews in focus group discussions. These terminologies have been vetted by Zambian social scientists and the two ethics boards in Zambia (UNZABREC and NHRA)

I think there is overreach re STIs throughout the protocol.

As above

Reviewer: 2

Dr. Sahai Burrowes, Touro University California

Comments to the Author:

I was happy to have the opportunity to review a manuscript on community-based schistosomiasis testing, as schistosomiasis causes great suffering but is woefully under-researched.

The protocol is ambitious, but I find the approach of simultaneously testing for multiple reproductive tract infections to be sound as their symptoms overlap, and I have found that misdiagnosis between them is common. It is also encouraging to see a study of the relationship between HIV, HPV, and schistosomiasis infection.

Thank you for your positive comment

I have minor suggestions listed below, but overall, the protocol is methodologically sound, reasonably well-written, and well-organized.

Editorial

The language could be cleaned up throughout the document. Minor typographical and grammatical errors are scattered throughout the document, and it will need thorough copy-editing before publication. In this editorial review, I suggest reducing the use of numbered lists as they are largely unnecessary in the text, and I don't believe they are commonly used in journal articles.

We have done copy-editing. Thank you

Methods

- Please state the formula used to calculate the study sample size and whether the clustering of responses by community block was considered when determining the sample size.

The sample size was determined on the basis of providing sufficient precision with which to estimate the sensitivity and specificity of self-sampling compared to clinic-based sampling for the detection of Sh. The formula used for this was the Clopper Pearson formula for calculating an exact 95% confidence

interval for a proportion. We targeted a minimum precision of +/-10% for estimating sensitivity within each district. Potential clustering of diagnostic test performance by community block was not considered

in this calculation.

We have expanded this in the protocol in lines (228-229) "we will be able to estimate the assumed target diagnostic sensitivity of 70% with a 95% CI of 62.3-76.6% overall (using the Clopper-Pearson formula to calculate an exact 95% confidence interval for this proportion)"

- One of the study aims was to empower women, but there is no information provided about the types of education that women receive before/during their participation in the study or the training that the community health workers will receive to educate women about these illnesses. A few more sentences about this would be nice to have. Educating women about the fact that their genital tract symptoms could mean several different diseases is important for future care-seeking and for reducing

stigma.

Thank you for this comment. We have added a few more sentences in the community-sensitisation section.

- The economic cost-effectiveness analysis seemed like an afterthought as it is discussed only cursorily. And I wonder if it shouldn't be removed from this protocol and written up separately in more detail. This protocol has many study outcomes and analyses regarding the shisto, HPV, and HPV infection. Perhaps the space in this manuscript would be better used to discuss those outcomes in more detail.

The cost-effectiveness analysis is certainly not an afterthought and we feel very strongly that it is an integral part of the study. In fact, it was very purposely added as a primary objective. An ambitious community-based project as the one we are conducting, needs a rigorous health economic analysis to establish if the strategy can be scaled-up and integrated within ongoing Ministry of Health screening programmes.

- Please cross-check the statistical analysis sections with the study design sections to ensure that the objectives/outcomes discussed are the same throughout the manuscript. For example, ART initiation is mentioned in the statistical analysis section but not as an outcome in the study design.

Done.

Tables and Figures

- For Figure 2, I would suggest adding the actions taken for women who are HIV-negative and HRHPV-positive to the flow chart so it is clear what follow-up care they receive. Are women who are HRHPV positive but who are not HIV infected referred to for VIA or other further care? VIA for HIV+ women is not listed in Figure 2, but it is mentioned in the text. This should be clarified.

Thank you for your comment. This is already included in the flowchart since the only difference between women with and without HIV is the age cut-off for the biopsy. This information is in the footnote of Figure 2. Also in lines 282-287.

- It's a bit repetitive, but I would have liked to have the location of sample collection included in Table 1. The location for the various sample collections is a little muddy in the text and would be easier to read in the table than in Figure 1.

We have edited Table 1 and added location of sample. We leave it to the editor to keep figure 1 which we quite like!

- Having a table or figure listing the various study objectives/outcome measures with their corresponding data collection and analysis methods and timing would be nice. To me, it seemed that study objectives were scattered throughout the document making it difficult to keep track of what outcomes the researchers would be measuring.

This would be in our view a little repetitive from the text since it is only a few lines stating the aims and objectives (Lines 172-179) but if the editor wants us to include an extra table we will be delighted to do so.